# Sampling to Distill: Knowledge Transfer from Open-World Data

Yuzheng Wang
Academy for Engineering and
Technology, Fudan University
Shanghai, China

Zhaoyu Chen
Academy for Engineering and
Technology, Fudan University
Shanghai, China

Jie Zhang
ETH Zurich
Zurich, Switzerland

Dingkang Yang
Zuhao Ge
Academy for Engineering and
Technology, Fudan University
Shanghai, China

Yang Liu
Siao Liu
Yunquan Sun
Academy for Engineering and
Technology, Fudan University
Shanghai, China

Wenqiang Zhang*
Lizhe Qi*
Engineering Research Center of AI &
Robotics, Ministry of Education,
Academy for Engineering and
Technology, Fudan University

## ABSTRACT

Data-Free Knowledge Distillation (DFKD) is a novel task that aims to train high-performance student models using only the pre-trained teacher network without original training data. Most of the existing DFKD methods rely heavily on additional generation modules to synthesize the substitution data resulting in high computational costs and ignoring the massive amounts of easily accessible, low-cost, unlabeled open-world data. Meanwhile, existing methods ignore the domain shift issue between the substitution data and the original data, resulting in knowledge from teachers not always trustworthy and structured knowledge from data becoming a crucial supplement. To tackle the issue, we propose a novel Open-world Data Sampling Distillation (ODSD) method for the DFKD task without the redundant generation process. First, we try to sample open-world data close to the original data's distribution by an adaptive sampling module and introduce a low-noise representation to alleviate the domain shift issue. Then, we build structured relationships of multiple data examples to exploit data knowledge through the student model itself and the teacher's structured representation. Extensive experiments on CIFAR-10, CIFAR-100, NYUv2, and ImageNet show that our ODSD method achieves state-of-the-art performance with lower FLOPs and parameters. Especially, we improve 1.50%-9.59% accuracy on the ImageNet dataset and avoid training the separate generator for each class.

## CCS CONCEPTS

• **Security and privacy** → **Data anonymization and sanitization**; **Database and storage security**; • **Computing methodologies** → *Reasoning about belief and knowledge.*

*Corresponding authors.

## KEYWORDS

Data-Free Knowledge Distillation, Open-World Unlabeled Data, Contrastive Learning, Relational Distillation

**ACM Reference Format:**
Yuzheng Wang, Zhaoyu Chen, Jie Zhang, Dingkang Yang, Zuhao Ge, Yang Liu, Siao Liu, Yunquan Sun, Wenqiang Zhang, and Lizhe Qi. 2024. Sampling to Distill: Knowledge Transfer from Open-World Data. In *Proceedings of the 32nd ACM International Conference on Multimedia (MM '24), October 28-November 1, 2024, Melbourne, VIC, AustraliaProceedings of the 32nd ACM International Conference on Multimedia (MM'24), October 28-November 1, 2024, Melbourne, Australia.* ACM, New York, NY, USA, 10 pages. https://doi.org/10.1145/3664647.3680618

## 1 INTRODUCTION

Deep learning has made refreshing progress in computer vision and multimedia fields [14, 20, 28, 30, 32, 37, 40, 42, 52, 56, 57, 60, 61, 68]. Despite the great success, large-scale models [9, 12, 26, 29, 31, 33, 41, 47, 51, 58, 59, 62] and unavailable privacy data [3, 46, 50, 53, 54] often impede the application of advanced technology on mobile devices. Therefore, model compression and data-free technology have become the key to breaking the bottleneck. To this end, Lopes *et al.* [34] propose Data-Free Knowledge Distillation (DFKD). In this process, knowledge is transferred from the cumbersome model to a small model that is more suitable for deployment without using the original training dataset. As a result, this widely applicable technology has gained much attention.

To replace unavailable private data, most existing data-free knowledge distillation methods rely on alternately training of the generator and student, called the generation-based method. However, these generation-based methods have many issues. First, their trained generators are abandoned after the students' training [6, 13, 16, 19, 36, 67]. The training of generators brings additional computational costs, especially for large datasets. For instance, a thousand generators are trained for the ImageNet dataset [11], which introduces more computational waste [15, 35]. Then, a serious domain shift issue exists between the generated substitution data and the original training data. Because the substitute data are composed of random noise transformation without supervision information and are highly susceptible to teacher preferences [54]. As a result, the efficiency and effectiveness of the generation-based methods are constrained, affecting student performance [2, 13, 39].

 

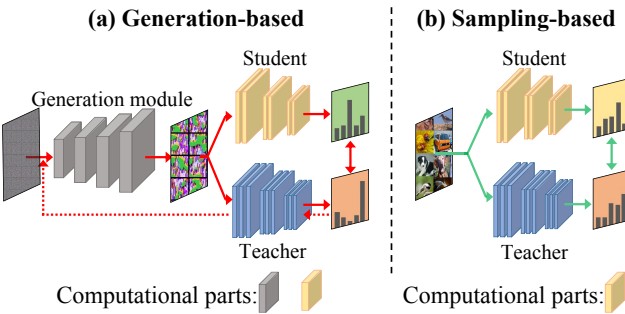

**Figure 1: Comparison of (a) generation-based and (b) sampling-based methods. The sampling-based process utilizes the open-world unlabeled data to distill the student network, so it does not need additional generation costs. At the same time, the extra knowledge in these unlabeled data enriches the knowledge representation from the teacher.**

Rather than relying on additional generation modules, Chen *et al.* [5] propose a sampling-based method to train the student network via unlabeled data without the generation calculations. Compared with generation-based methods, sampling-based methods can avoid the training cost of generators, thus improving algorithm efficiency. The comparison of the two methods is shown in Figure 1. Meanwhile, they try to reduce label noise by updating the learnable noise matrix, but the noise matrix's computational costs are expensive. More restrictedly, their sampling method relies on the strict confidence ranking and does not consider the data domain similarity issue (We discuss the distribution similarity between sampled data and original data in detail in Section 4.4). In addition, the existing generation-based and sampling-based methods can be summarized as simple imitation learning, *i.e.*, the student mimics the output of a particular data example represented by the teacher [37, 53, 64]. Therefore, these methods do not adequately utilize the implicit relationship among multiple data examples, which leads to the lack of effective knowledge expression in the distillation process.

Based on the above observations, we construct a sampling-based method to sample helpful data from easily accessible, low-cost, unlabeled open-world data, avoiding unnecessary computational costs. In addition, we propose two aspects of customized optimization. (**i**) To cope with the domain shift issue between the open-world and original data, we preferentially try to sample data with a similar distribution to the original data domain to reduce the shifts and design a low-noise knowledge representation learning module to suppress the interference of label noise from the teacher model. (**ii**) To explore the data knowledge adequately, we set up a structured representation of unlabeled data to enable the student to learn the implicit knowledge among multiple data examples. As a result, the student can learn from carefully sampled unlabeled data instead of relying on the teacher. At the same time, to explore an effective distillation process, we introduce a contrastive structured relationship between the teacher and student. The student can make better progress through the structured prediction of the teacher network.

In this paper, we consider a solution to the DFKD task that does not require additional generation costs. On the one hand,

we look forward to the solution to data domain shifts from both data source and distillation methods. On the other hand, we try to explore an effectively structured knowledge representation method to deal with the missing supervision information and the training difficulties in the DFKD scenes. Therefore, we propose an Open-world Data Sampling Distillation (ODSD) method, which includes Adaptive Prototype Sampling (APS) and Denoising Contrastive Relational Distillation (DCRD) modules. Specifically, the primary contributions and experiments are summarized as follows:

- We propose a sampling-based method with the unlabeled open-world data. The method does not require additional training of one or more generator models, thus avoiding unnecessary computational costs and model parameters.
- During the sampling process, considering the domain shifts between the unlabeled data and the original data, we propose an Adaptive Prototype Sampling (APS) module to obtain data closer to the original data distribution.
- During the distillation process, we propose a Denoising Contrastive Relational Distillation (DCRD) module to suppress label noise and exploit knowledge from data and the teacher more adequately by building structured relationships.
- The proposed method achieves state-of-the-art performance with lower FLOPs, improves the effectiveness of the sampling process, and alleviates the distribution shift between the unlabeled data and the original data.

## 2 RELATED WORK

### 2.1 Data-Free Knowledge Distillation

Data-free knowledge distillation aims to train lightweight models when the original data are unavailable. Therefore, the substitute data are indispensable to help transfer knowledge from the cumbersome teacher to the flexible student. According to the source of these data, existing methods are divided into generation-based and sampling-based methods.

**Generation-based Methods.** The generation-based methods depend on the generation module to synthesize the substitute data. Lopes *et al.* [34] propose the first generation-based DFKD method, which uses the data means to fit the training data. Due to the weak generation ability, it can only be used on a simple dataset such as the MNIST dataset. The following methods combine the Generative Adversarial Networks (GANs) to generate more authentic and reliable data. Chen *et al.* [6] firstly put the idea into practice and define an information entropy loss to increase the diversity of data. However, this method relies on a long training time and a large batch size. Fang *et al.* [16] suggest forcing the generator to synthesize images that do not match between the two networks to enhance the training effect. Hao *et al.* [19] suggest using multiple pre-trained teachers to help the student, which leads to additional computational costs. Do *et al.* [13] propose a momentum adversarial distillation method to help the student recall past knowledge and prevent the student from adapting too quickly to new generator updates. The same domain typically shares some reusable patterns, so Fang *et al.* [15] introduce the sharing of local features of the generated graph, which speeds up the generation process. Since the generation quality is still not guaranteed, some methods spend extra computational costs on gradient inversion to synthesize more

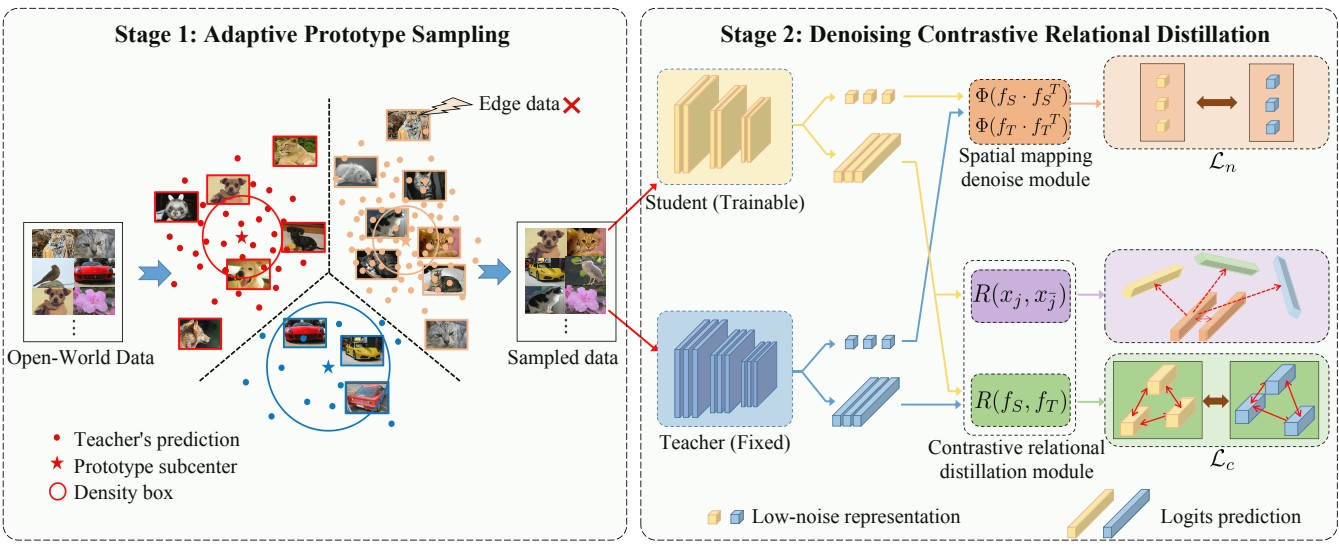

**Figure 2: The pipeline of our proposed ODSD. First, all open-world unlabeled data passes through adaptive prototype sampling so that the substitute dataset resembles the distribution of the original data. Then, based on these data, the student can make progress through low-noise information representation, data knowledge mining, and structured knowledge from the teacher.**

realistic data [17, 65]. In addition, Choi *et al.* [10] combine DFKD with other compression technologies and achieve encouraging performance. However, generation-based DFKD methods generate a large number of additional calculation costs in generation modules, while these modules will be discarded after students' training [5].
**Sampling-based Methods.** To train the student more exclusively, Chen *et al.* [5] propose to sample unlabeled data to replace the unavailable data without the generation module. Firstly, they use a strict confidence ranking to sample unlabeled data. Then, they propose a simple distillation method with a learnable adaptive matrix. Despite no additional generating costs and promoting encouraging results, their method ignores the intra-class relationships of multiple unlabeled data. Simultaneously, the simple strict confidence causes more data to be sampled for simple classes, leading to imbalanced data classes. In addition, their proposed distillation method is relatively simple and lacks structured relationship expression, which limits the student's performance.

## 2.2 Contrastive Learning

Contrastive learning makes the model's training efficient by learning the data differences [63]. The unsupervised training pipeline usually requires storing negative data by a memory bank [55], large dictionaries [21], or a large batch size [8]. Even it requires a lot of computation costs, *e.g.*, additional normalization [18], and network update operations [4]. The high storage and computing costs seriously reduce knowledge distillation efficiency. But at the same time, this idea of mining knowledge in unlabeled data may be helpful for the student's learning. Due to such technical conflicts, there are few methods to combine knowledge distillation and contrastive learning in the past perfectly. As a rare attempt, Tian *et al.* [48] propose a contrastive data-based distillation method by updating a large memory bank. However, data quality cannot be guaranteed

for data-free knowledge distillation, and data domain shifts are intractable, making the above process challenging.

In this work, we attempt to explore additional knowledge from both the data and the teacher. Therefore, we further stimulate students' learning ability by using the internal relationship of unlabeled data and constructing a structured contrastive relationship. To our best knowledge, this is the first combination of data-free knowledge distillation and contrastive learning at a low cost during the distillation process, which achieves an unexpected effect.

## 3 METHODOLOGY

### 3.1 Overview

Our pipeline includes two stages: **(i)** unlabeled data sampling and **(ii)** distillation training, as shown in Figure 2. For the sampling stage, we sample unlabeled data by an adaptive sampling mechanism to obtain data closer to the original distribution. For the distillation stage, the student learns the knowledge representation after denoising through a spatial mapping denoise module. Further, we mine more profound knowledge of the unlabeled data and build the structured relational distillation to help the student gain better performance. The complete algorithm is shown in Algorithm 1.

### 3.2 Adaptive Prototype Sampling

The unlabeled data and the original data are distributed differently in many cases. To obtain the substitution data with a more similar distribution to the original data from the specific unlabeled dataset, we propose an Adaptive Prototype Sampling (APS) module, which considers the teacher's familiarity with the data, excludes misclassified offset noisy data, and focuses on the class balance of the sampled data. Based on these, we design three score indicators

---

**Algorithm 1** The proposed ODSD algorithm.

---

**Input:** A frozen teacher network $f_T$, an unlabeled open-world dataset $X_U$, and the target number of sampled data $M$.

1: **Module 1: Adaptive pototype sampling**
2: **for** unlabeled data $x_i$ in $X_U$ **do**
3:     Classify teacher predictions $p_i$ as $\rho_{i,c} = p_i \in c$;
4:     Calculate confidence probability: $\tilde{p}_i = \sigma(p_i)$
5:     Cluster the prediction vector as the prototypes $\mu_{c,k}$.
6: **end for**
7: **for** Prototypes $\mu_{c,k}$ in class $c$ **do**
8:     Obtain prototype similarity: $\tilde{o}_i = \cos(\rho_{i,c}, \mu_{c,k})_{k=1}^K$;
9:     Calculate intra-class outliers mean: $u_c = \frac{1}{n_c} \sum_{p_i \in c} \tilde{o}_i$;
10:     Calculate the density score $D_c = \frac{\sqrt{u_c}}{\log_e (n_c + C)}$.
11: **end for**
12: Calculate sampling score: $S = \frac{\tilde{p}_i}{|\max\{\tilde{p}\}|} - \frac{\tilde{o}_i}{|\max\{\tilde{o}\}|} + \frac{D_c}{|\max\{D\}|}$
13: Sample top-$M$ data with the highest score as $X_A$.
14: **Module 2: Denoising contrastive relational distillation.**
15: **for** $i$ in number of epochs **do**
16:     **for** training data $x$ in $X_A$ **do**
17:         Calculate $\mathcal{L}_{total}$ as Eq.8 and update the student $f_S$.
18:     **end for**
19: **end for**
**Output:** The trained student $f_S$ and a reusable sampling list $L$ of the teacher $f_T$ on dataset $X_U$.

---

to evaluate the effectiveness of the unlabeled data for student training corresponding to the above three aspects, including the data confidence score, the data outlier score, and the class density score.

**(a) Data Confidence Score.** The teacher provides the prediction logits $P = [p_1, \ldots, p_n] \in \mathbb{R}^{n \times C}$ on the unlabeled dataset $\{x_0, \ldots, x_n\}$, where $p_i$ denotes the prediction for the $i$-th sample satisfying $p_i \in \mathbb{R}^{1 \times C}$. $n$ denotes the number of data, and $C$ denotes the number of classes. Then the prediction is converted into the probability of the unified scale as $\tilde{p}_i = \sigma(p_i)$, where $\sigma$ denotes the softmax layer and $\tilde{p}_i$ denotes the confidence probability corresponding to the predicted result class. Therefore, $\tilde{p} = [\tilde{p}_1, \ldots, \tilde{p}_n]$ represents the confidence of each data in the unlabeled dataset. We choose the largest $\max\{\tilde{p}\}$ for normalization. The confidence score of $i$-th sample $x_i$ can be calculated as: $sc_i = \frac{\tilde{p}_i}{|\max\{\tilde{p}\}|}$.

**(b) Data Outlier Score.** The data distribution of the substitution data and the original data is different. Therefore, the confusing edge data should be excluded, *i.e.*, the data with different distributions but also predicted as the same target class. For example, a tiger is predicted as the class of cat, as shown in the orange part of Stage 1 in Figure 2. We first separate the teacher predictions according to the predicted classes as $\rho_{i,c} = p_i \in c$. For each class, $\rho_{i,c}$ is clustered [25] to explore the intra-class relationships through $k$ layering as $\mu_{c,k}$. Then the prediction features for the whole unlabeled dataset can be expressed as a group of $CK$ prototypes as $\{\mu_{c,k} \in \mathbb{R}^{1 \times C}\}_{c,k=1}^{C,K}$, where $c$ denotes the $c$-th class, and $K$ denotes the hyperparameter representing the number of prototypes for each class. The prototype centers of the $c$-th class can be expressed as $\{\mu_{c,k}\}_{k=1}^K$. The outlier of each unlabeled data $x_i$ can be calculated with the sum of the

prototype centers of its class as $\tilde{o}_i = \sum_{k=1}^K \cos(\rho_{i,c}, \mu_{c,k})$, where cos denotes the cosine similarity metric. Similar to the confidence score, we select the maximum value $\max\{\tilde{o}\}$ for normalization. As a result, the outlier score can be calculated as: $so_i = \frac{\tilde{o}_i}{|\max\{\tilde{o}\}|}$.

**(c) Class Density Score.** To help the student learn various classes effectively, we calculate the class density for the class balance of the sampled data. As shown in Stage 1 of Figure 2, we increase the sampling range for classes with sparse data (the blue part) while we reduce the sampling range for classes with redundant data (the orange part). Based on this, we first separate the above intra-class outliers $\tilde{o}_i$ of all data by their predicted classes. The outliers mean value of each class can be calculated as $u_c = \frac{1}{n_c} \sum_{p_i \in c} \tilde{o}_i$, where $n_c$ is the number of the data predicted as $c$-th class. Therefore, the Dcluster parameter $D_c$ can be calculated as: $D_c = \frac{\sqrt{u_c}}{\log_e (n_c + C)}$, which reflects the data density predicted to be $c$-th class. The introduction of a constant $C$ (the number of classes) helps the numerical stability when the available unlabeled data is small while having little effect on the results when the amount of data is sufficient (under normal conditions). After selecting the maximum value $\max\{D\}$ for normalization, the density score of each data can be calculated according to the predicted class as $sd_i = \frac{D_c}{|\max\{D\}|}$, when $\arg\max(p_i) = c$.

Finally, we calculate the total score as $S_{total} = sc_i - so_i + sd_i$. Based on this, the data closer to the distribution of the original data domain are sampled, which can help the student learn better. The quantitative analysis is shown in Table 7.

## 3.3 Denoising Contrastive Relational Distillation

After obtaining the high score data, the distillation process can be carried out. We denote $f_T$ and $f_S$ as the teacher and student networks and denote $x$ as the data in sampled set $X_A$. According to the definition [23], the knowledge distillation loss is calculated as:

$$\mathcal{L}_{KD} = \sum_{x \in X_A} D_{KL}(f_T(x)/\tau_{kd}, f_S(x)/\tau_{kd}), \tag{1}$$

where $D_{KL}$ is the Kullback-Leibler divergence, and $\tau_{kd}$ is the distillation temperature. $\mathcal{L}_{KD}$ allows the student to imitate the teacher's output. However, the main challenge is the distribution differences between the substitute and original data domains, leading to label noise interference. Simultaneously, the ground-truth labels are unavailable, so correct information supervision is missing. Therefore, we propose a Denoising Contrastive Relational Distillation (DCRD) module, which includes a spatial mapping denoise component and a contrastive relationship representation component to help the student get better performance and mitigate label noise.

**Spatial Mapping Denoise.** The distribution in the unlabeled data differs from the unavailable original data, which indicates the inevitable label noise. Inspired by manifold learning [44], low dimensional information representation contains purer knowledge with less noise interference [1]. Here, we utilize a low-dimensional spatial mapping denoise component to help the student learn low-noise knowledge representation. Based on this, we perform eigendecomposition $\Phi$ on the teacher's prediction and its transposed product

matrix [24]. According to the distance invariance, the autocorrelation matrix $d_{ij}^2$ in a mini-batch as:

$$\sum_i^N \sum_j^N d_{ij}^2 = 2N \cdot tr(Z_t Z_t^T), \tag{2}$$

where $N$ denotes the batch size, and $tr(\cdot)$ denotes the trace of a matrix. $Z_t$ is the low-dimensional spatial vector representation from the teacher calculated as $\Phi(f_T(x) \cdot f_T^T(x)) = Z_t = V_t \Lambda_t^{1/2}$, where $V_t$ is the eigenvalue, and $\Lambda_t$ is the eigenvector. Similarly, we can get the student predictions of low-dimensional representation as $Z_s$. Then, we set up a distillation loss to correct the impact of label noise by the spatial mapping of the two networks. The spatial mapping denoise distillation loss is calculated as:

$$\mathcal{L}_n = \ell_h(\Phi(f_T(x) \cdot f_T^T(x)), \Phi(f_S(x) \cdot f_S^T(x))) = \ell_h(Z_t, Z_s), \tag{3}$$

where $\ell_h(\cdot, \cdot)$ denotes the Huber loss.

**Contrastive Relational Distillation.** The missing supervision information limits the student's performance. It is indispensable to adequately mine the knowledge in unlabeled data to compensate for the lack of information. To avoid a single imitation of a particular data example, we build two kinds of structured relationships to mine knowledge from the data and the teacher.

Firstly, the student can adequately explore the structured relation among multiple unlabeled data by learning the instance invariant. $x_i, x_j$ are the different data in a mini-batch. We calculate the prediction difference between data as follows:

$$\ell_s^{x_i x_j} = \frac{\cos(f_S(x_i), f_S(x_j))/\tau}{\sum_{k=1, k \neq i}^{2N} \cos(f_S(x_i), f_S(x_k))/\tau}, \tag{4}$$

where $\tau_1$ denotes contrastive temperature. Next, we can calculate the consistency instance discrimination loss as:

$$\mathcal{L}_{c1} = -\frac{1}{N} \sum_{j=1}^{N} \log \ell_s^{x_j \bar{x}_j}, \tag{5}$$

where $\bar{x}_j$ denotes the strong data augmentation of $x_j$. This unsupervised method is especially effective when the teacher makes wrong results.

Secondly, we construct a structured contrastive relationship between the teacher and student, which promotes consistent learning between the teacher and student. The structured knowledge transfer process is calculated as:

$$\ell_{ts}^{x_i'} = \frac{\cos(f_T(x_i'), f_S(x_i'))/\tau}{\sum_{k=1, k \neq i}^{4N} \cos(f_T(x_i'), f_S(x_k'))/\tau}, \tag{6}$$

where $x' = x \cup \bar{x}$ denotes the set of the sampled data before and after strong data augmentation. And $x'$ contains $2N$ samples for each batch. We calculate the teacher-student consistency loss as:

$$\mathcal{L}_{c2} = -\frac{1}{2N} \sum_{j=1}^{2N} \log \ell_{ts}^{x_j'}. \tag{7}$$

The student can obtain better learning performance through the mixed structured and consistent relationship learning between the two networks. Then, the contrastive relational distillation loss is

**Table 1: Illustration of original private data and their corresponding substitute open-world datasets.**

| Original data | CIFAR | ImageNet | NYUv2 |
|---|---|---|---|
| Unlabeled data | ImageNet | Flickr1M | ImageNet |

$\mathcal{L}_c = \mathcal{L}_{c1} + \mathcal{L}_{c2}$. Finally, we can get the total denoising contrastive relational distillation loss as:

$$\mathcal{L}_{total} = \mathcal{L}_{KD} + \lambda_1 \cdot \mathcal{L}_n + \lambda_2 \cdot \mathcal{L}_c, \tag{8}$$

where $\lambda_1, \lambda_2$ are the trade-off parameters for training losses.

## 4 EXPERIMENTS

### 4.1 Experimental Settings

**Datasets.** We evaluate the proposed ODSD method for the classification and semantic segmentation tasks. For classification, we evaluate it on widely used datasets: $32 \times 32$ CIFAR-10, CIFAR-100 [27], and $224 \times 224$ ImageNet [11]. For semantic segmentation, we evaluate the proposed method on $128 \times 128$ NYUv2 dataset [45]. Besides, the corresponding open-world datasets are shown in Table 1, which is the same as DFND [5] for a fair comparison.

**Implementation Details.** The proposed model is implemented in PyTorch [38] and trained with RTX 3090 GPUs. For the CIFAR-10 and CIFAR-100 datasets, we conduct five sets of backbone combinations, set two groups of different numbers of sampled samples (150k or 600k), and train the students for 200 epochs. For the ImageNet dataset, we conduct three sets of backbone combinations and train the students for 200 epochs. The number of sampled samples is 600k. For the NYUv2 dataset, the DeeplabV3 [7] is used as the model architecture followed previous work. The teacher uses ResNet-50 [22] as the backbone, and the student uses mobilenetv2 [43]. We sample 200k unlabeled samples and train the student for 20 epochs. For the above datasets, we set $\tau_{kd}$ as 4 to be the same as other distillation methods and set $\tau$ as 0.5 to be the same as [8]. Besides, we set $\lambda_1$ as 10 and $\lambda_2$ as 0.5, use the SGD optimizer with momentum as 0.9, weight decay as $5 \times 10^{-4}$, the batch size $N$ as 64, and cosine annealing learning rate with an initial value of 0.025.

**Baselines.** We compare generation-based and sampling-based DFKD methods, including DeepInv [65], CMI [17], DAFL [6], ZSKT [36], DFED [19], DFQ [10], Fast [15], MAD [13], DFD [35], KAKR [39], SpaceshipNet [66], DFAD [16], and DFND [5].

### 4.2 Performance Comparison

To evaluate the effectiveness of our ODSD, we compare it with SOTA DFKD methods regarding the student's performance, the effectiveness of the sampling method, and training costs.

**Experiments on CIFAR-10 and CIFAR-100.** We first verify the proposed method on the CIFAR-10 and CIFAR-100 [27]. The baseline "*Teacher*" and "*Student*" means to use the corresponding backbones of the teacher or student for direct training with the original training data, and "*KD*" represents distilling the student network with the original training data. Generation-based methods include training additional generators and calculating model gradient inversion. Sampling-based methods use the unlabeled ImageNet dataset. We

**Table 2: Student accuracy (%) on CIFAR datasets. Bold and underline numbers denote the best and the second best results.**

| Dataset | Method | Type | ResNet-34 ResNet-18 | VGG-11 ResNet-18 | WRN40-2 WRN16-1 | WRN40-2 WRN40-1 | WRN40-2 WRN16-2 |
|---|---|---|---|---|---|---|---|
| CIFAR-10 | Teacher | - | 95.70 | 92.25 | 94.87 | 94.87 | 94.87 |
| | Student | | 95.20 | 95.20 | 91.12 | 93.94 | 93.95 |
| | KD | | 95.58 | 94.96 | 92.23 | 94.45 | 94.52 |
| | DeepInv [65] | Generation | 93.26 | 90.36 | 83.04 | 86.85 | 89.72 |
| | CMI [17] | | 94.84 | 91.13 | 90.01 | 92.78 | 92.52 |
| | DAFL [6] | | 92.22 | 81.10 | 65.71 | 81.33 | 81.55 |
| | ZSKT [36] | | 93.32 | 89.46 | 83.74 | 86.07 | 89.66 |
| | DFED [19] | | - | - | 87.37 | 92.68 | 92.41 |
| | DFQ [10] | | 94.61 | 90.84 | 86.14 | 91.69 | 92.01 |
| | Fast [15] | | 94.05 | 90.53 | 89.29 | 92.51 | 92.45 |
| | MAD [13] | | 94.90 | - | - | - | 92.64 |
| | KAKR_MB [39] | | 93.73 | - | - | - | - |
| | KAKR_GR [39] | | 94.02 | - | - | - | - |
| | SpaceshipNet [66] | | 95.39 | 92.27 | 90.38 | 93.56 | 93.25 |
| | DFND_150$k$ [5] | Sampling | 94.18 | 91.77 | 87.95 | 92.56 | 92.02 |
| | DFND_600$k$ [5] | | 95.36 | 91.86 | 90.26 | 93.33 | 93.11 |
| | ODSD_150$k$ | | 95.05 | 92.02 | 89.14 | 92.94 | 92.34 |
| | ODSD_600$k$ | | **95.70** | **92.55** | **91.53** | **94.31** | **94.02** |
| CIFAR-100 | Teacher | - | 78.05 | 71.32 | 75.83 | 75.83 | 75.83 |
| | Student | | 77.10 | 77.10 | 65.31 | 72.19 | 73.56 |
| | KD | | 77.87 | 75.07 | 64.06 | 68.58 | 70.79 |
| | DeepInv [65] | Generation | 61.32 | 54.13 | 53.77 | 61.33 | 61.34 |
| | CMI [17] | | 77.04 | 70.56 | 57.91 | 68.88 | 68.75 |
| | DAFL [6] | | 74.47 | 54.16 | 20.88 | 42.83 | 43.70 |
| | ZSKT [36] | | 67.74 | 54.31 | 36.66 | 53.60 | 54.59 |
| | DFED [19] | | - | - | 41.06 | 60.96 | 60.79 |
| | DFQ [10] | | 77.01 | 66.21 | 51.27 | 54.43 | 64.79 |
| | Fast [15] | | 74.34 | 67.44 | 54.02 | 63.91 | 65.12 |
| | MAD [13] | | 77.31 | - | - | - | 64.05 |
| | KAKR_MB [39] | | 77.11 | - | - | - | - |
| | KAKR_GR [39] | | 77.21 | - | - | - | - |
| | SpaceshipNet [66] | | 77.41 | 71.41 | 58.06 | 68.78 | 69.95 |
| | DFND_150$k$ [5] | Sampling | 74.20 | 69.31 | 58.55 | 68.54 | 69.26 |
| | DFND_600$k$ [5] | | 74.42 | 68.97 | 59.02 | 69.39 | 69.85 |
| | ODSD_150$k$ | | 77.90 | 72.24 | 60.55 | 71.66 | 72.42 |
| | ODSD_600$k$ | | **78.45** | **72.71** | **60.57** | **72.71** | **73.20** |

reproduce the DFND using the unified teacher models, and the result is slightly higher than the original paper.

As shown in Table 2, our ODSD has achieved the best results on each baseline. Under most baseline settings, ODSD brings gains of 1% or even higher than the SOTA methods, even though students' accuracy is very close to their teachers. In particular, the students of our ODSD outperform the teachers on some baselines. As far as we know, it is the first DFKD method to achieve such performance. The main reasons for its breakthrough in analyzing the algorithm's performance come from three aspects. First, our data sampling method comprehensively analyzes the intra-class relationships in the unlabeled data, excluding the difficult edge data and significant distribution differences data. At the same time, the number of data in each class is relatively more balanced, which is conducive to all kinds of balanced learning compared with other sampling methods. Second, our knowledge distillation method considers the representation of low-dimensional and low-noise information and expands

the representation of knowledge through data augmentation. The structured relationship distillation method helps the student effectively learn knowledge from both multiple data and its teacher. Finally, the knowledge of our ODSD does not entirely come from the teacher but also the consistency and differentiated representation learning of unlabeled data, which is helpful when the teacher makes mistakes. The previous methods ignore the in-depth mining of data knowledge, decreasing students' performance.

**Experiments on ImageNet.** We conduct experiments on a large-scale ImageNet dataset to further verify the effectiveness. Due to the larger image size, it is challenging to effectively synthesize training data for most generation-based methods. Generation-based methods train 1,000 generators (one generator for one class), resulting in a large amount of computational costs. In this case, our sampling method reduces the computational costs more significantly. We set up three baselines to compare the performance of our method with the SOTA methods. Table 3 reports the experimental results.

**Table 3: Student accuracy (%) on ImageNet dataset.**

| Method | Type | ResNet-50 ResNet-18 | ResNet-50 ResNet-50 | ResNet-50 MobileNetv2 |
|--------|------|-----------|-----------|-----------|
| Teacher | | 75.59 | 75.59 | 75.59 |
| Student | - | 68.93 | 75.59 | 63.97 |
| KD | | 68.10 | 74.76 | 61.67 |
| DFD [35] | | 54.66 | 69.75 | 43.15 |
| DeepInv$_{2k}$ [65] | Generation | - | 68.00 | - |
| Fast$_{50}$ [15] | | 53.45 | 68.61 | 43.02 |
| DFND [5] | Sampling | 42.82 | 59.03 | 16.03 |
| **ODSD** | | **58.24** | **71.25** | **52.74** |

**Table 4: Total FLOPs and params in DFKD methods.**

| Method | DeepInv | CMI | DAFL | ZSKT | DFQ | DFND | **ODSD** |
|--------|---------|-----|------|------|-----|------|------|
| FLOPs | 4.36G | 4.56G | 0.67G | 0.67G | 0.79G | 0.56G | 0.56G |
| params | 11.7M | 12.8M | 12.8M | 12.8M | 17.5M | 11.7M | 11.7M |

**Table 5: APS compared with the SOTA sampling method.**

| Sampling methods | KD | Method DFND | **ODSD** |
|------------------|-----|------|------|
| Random | 76.85 | 73.15 | 76.43 |
| DFND | 76.67 | 73.68 | 77.40 |
| **APS** | **77.27** | **73.89** | **77.90** |

Our ODSD still achieves several percentage points increase compared with other SOTA methods, especially in the cross-backbones situation (9.59%). Due to the lack of structured knowledge representation, the DNFD algorithm performs poorly on the large-scale dataset. Comparing DFND and ODSD, our structured framework improves the overall understanding ability of the student.

**Comparison of Training Costs.** We further calculate the total floating point operations (FLOPs) and parameters (params) required by various DFKD algorithms, as shown in Table 4. Our method only needs training costs and params of the student network without additional generation modules. Our sampling process introduces 256.78 seconds for sample selection ($K = 5$) on the CIFAR100 with a single RTX 3090 GPU (The teacher uses the *ResNet-34*) while the fastest generation-based method ZSKT also takes 1.54 hours to synthesize data. These generation modules will be discarded after student training, which causes a waste of computing power.

**Comparison of Data Sampling Efficiency.** To verify the sampling mechanism's effectiveness, we compare our APS method with the current SOTA unlabeled data sampling method DFND [5]. Three data sampling methods (random sampling, DFND sampling, and our proposed APS) are set on three different distillation algorithms, including: KD [23], DFND [5], and our ODSD method. Table 5 reports the results. For KD, we use the sampled data instead of the original generated data with $\mathcal{L}_{KD}$ distillation loss. From the result, this setting is competitive, even better than the distillation loss of DFND. For DFND, we reproduce it with open-source codes and keep the original training strategy unchanged. We find the performance of

**Table 6: Segmentation results on NYUv2 dataset.**

| Algorithm | Teacher | Student | DAFL | DFAD | Fast | DFND | **ODSD** |
|-----------|---------|---------|------|------|------|------|------|
| mIoU | 0.517 | 0.375 | 0.105 | 0.364 | 0.366 | 0.378 | **0.397** |

**Table 7: Diagnostic studies of the proposed method.**

| ID | Training objective $\mathcal{L}$ Setting | Accuracy (%) 50k | 150k | ID | Data sampling scores $S$ Setting | Accuracy (%) 50k | 150k |
|----|---------|------|------|----|---------|------|------|
| (1) | ours | **75.26** | **77.90** | (5) | ours | **75.26** | **77.90** |
| (2) | w/o $\mathcal{L}_n$ | 74.82 | 77.71 | (6) | w/o $sc_i$ | 73.96 | 77.04 |
| (3) | w/o $\mathcal{L}_c$ | 74.71 | 77.58 | (7) | w/o $so_i$ | 68.07 | 76.67 |
| (4) | w/o $\mathcal{L}_n, \mathcal{L}_c$ | 74.39 | 77.27 | (8) | w/o $sd_i$ | 70.24 | 76.59 |

the DFND sampling method is unstable, which causes it to be lower than random sometimes. For ODSD, we use the distillation loss in Equation (8). Our proposed sampling method achieves the best performance in all three benchmarks and significantly improves performance. By comprehensively considering the data confidence, the data outliers, and the class density, our ODSD can more fully mine intra-class relationships of the unlabeled data. As a result, the sampled data are more helpful for subsequent student learning.

**Experiments about Semantic Segmentation.** In addition to image classification tasks, our algorithm can also effectively solve the problem of DFKD in image semantic segmentation on the NYUv2 dataset. Mean Intersection over Union (mIoU) is set as the segmentation evaluation metric. No generation module is defined for our method, and other settings are the same as DFAD [16]. Table 6 shows segmentation results on the NYUv2 dataset. Our ODSD also achieves the best performance. Besides, we visualize the segmentation results of different networks to get more convincing results as shown in Figure 3. "*Input*" and "*Ground Truth*" represent the input test data and their corresponding real labels. Most data-free distillation algorithms hide the code of the segmentation part, so it is not easy to make a visual comparison. Here, we choose DFAD as the baseline algorithm of visualization. Our proposed ODSD algorithm achieves better segmentation results than DFAD, especially for object contour segmentation. The slight noise around the contour is effectively suppressed. Further, through in-depth mining the knowledge from the data and teacher, our student have gained better understanding ability.

### 4.3 Diagnostic Experiment

We conduct the diagnostic studies on the CIFAR-100 dataset and use ResNet-34 and ResNet-18 as the teacher's and student's backbones with 50k and 150k sampled data.

**Distillation Training Objective.** We first investigate our overall training objective (cf. Equation (8)). As shown in the experiments (1-4) of Table 7, the model with $\mathcal{L}_{KD}$ alone achieves accuracy scores of 74.39% and 77.27% on 50k and 150k data sampling settings. Adding $\mathcal{L}_n$ or $\mathcal{L}_c$ individually brings gains (*i.e.*, **0.32%, 0.31%/ 0.43%, 0.44%**), indicating the effectiveness of our proposed distillation method. Our method performs better with

| Input | Ground Truth | Teacher | DFAD | ODSD (ours) |
|---|---|---|---|---|

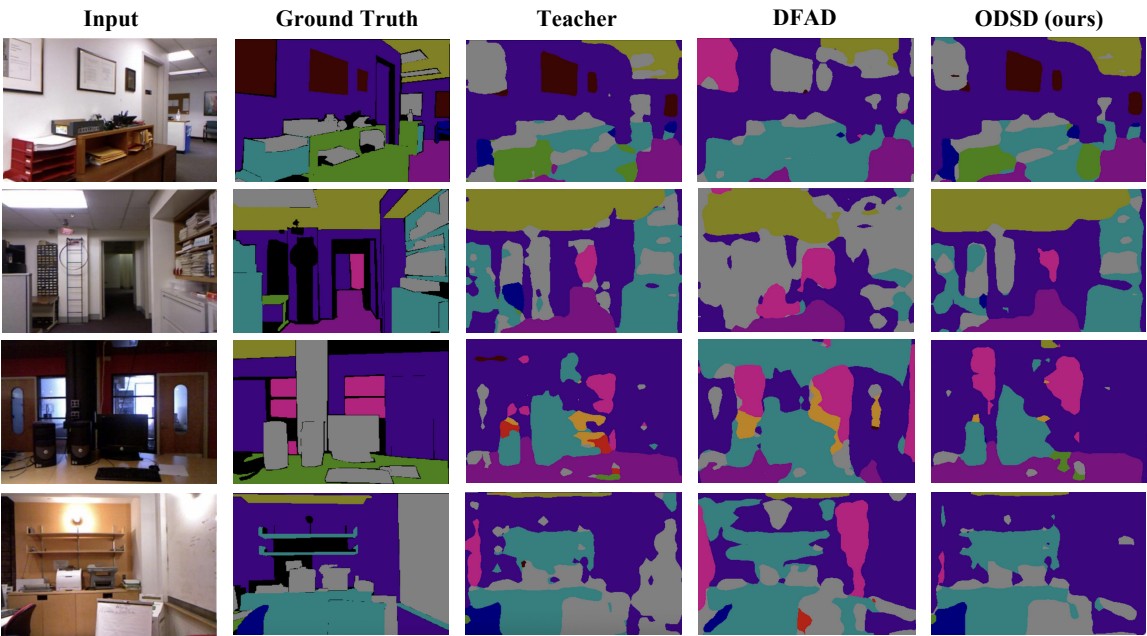

**Figure 3: Visualization segmentation results on the NYUv2 dataset.**

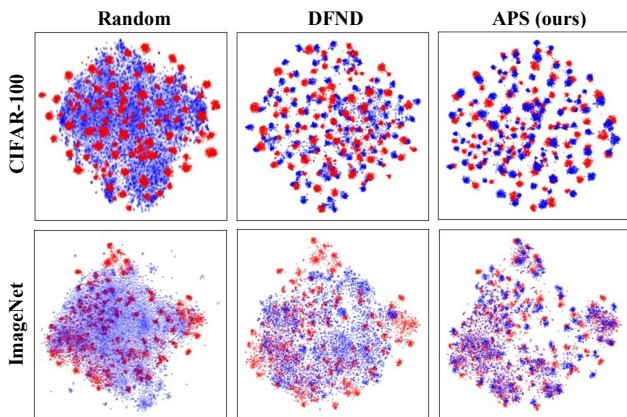

**Figure 4: t-SNE visualization of the data distributions on CIFAR-100 and ImageNet datasets. Red dots denote original domain data, while blue dots denote unlabeled sampled data.**

**75.26%** and **77.90%**. Therefore, the proposed training objectives are effective and can help the student gain better performance.

**Data Sampling Scores.** We further verify the effectiveness of the three sampling scores in section 3.2. Using all scores, the model can achieve the best performance with **75.26%** and **77.90%** accuracy shown in experiments (5-8) of Table 7. When the confidence score $sc_i$ is abandoned, the familiarity of the teacher network with the sampled data decreases, reducing the amount of adequate information contained in the data. Without the outlier score $so_i$, the lack of modeling of the intra-class relationship of the data to be sampled leads to increased data distribution difference between the substitute data domain and the original data domain. Further,

the class density score $sd_i$ can measure the number of data in each class and maintain the balance of the sampled data. In summary, all three score indicators can help students perform better.

## 4.4 Visualization

To verify the distribution similarity between the sampled data and the original data of our APS sampling method and the DFND sampling method, we use t-SNE [49] to visualize the data feature distribution. The teacher uses ResNet-34 on the CIFAR-100 and ResNet-50 on the ImageNet. For both datasets, we reserve 100 classes from validation data. In addition, we also visualize the distribution of data obtained by random sampling as a baseline reference. Figure 4 shows the data distribution results of different sampling methods. Our clustering results are closer to the extracted features of the original data. For the more complex ImageNet, this advantage is further amplified. Reducing the distribution difference between sampled and original data helps reduce data label noise, which is the key for the student to perform well.

## 5 CONCLUSION

Most existing data-free knowledge distillation methods rely heavily on additional generation modules, bringing additional computational costs. Meanwhile, these methods disregard the domain shifts issue and ignore the data knowledge. This paper proposes an Open-world Data Sampling Distillation method. We sample unlabeled data with a similar distribution to the original data and introduce low-noise representation learning to cope with domain shifts. To explore the data knowledge adequately, we design a structured knowledge representation. Comprehensive experiments illustrate the effectiveness of the proposed method, which achieves state-of-the-art performance on various benchmarks.

## ACKNOWLEDGEMENTS

This work is supported by the Shanghai Engineering Research Center of AI & Robotics, Fudan University, China, the Engineering Research Center of AI & Robotics, Ministry of Education, China, and the Green Ecological Smart Technology School-Enterprise Joint Research Center.

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
