# OpenReview forum: "Sampling to Distill: Knowledge Transfer from Open-World Data"
_acmmm.org/ACMMM/2024/Conference — MM2024 Poster_

### Official Review · Reviewer_vm8r · 2024-05-23

**Rating:** 4
**Confidence:** 3

**Summary:**

This paper proposes a novel Open-world Data Sampling Distillation (ODSD) method for the DFKD task, which samples open-world unlabeled data to replace the fake data via an adaptive sampling module.  Extensive experiments demonstrate the effectiveness of ODSD.

**Strengths:**

1) ODSD utilizes the open-world unlabeled data to perform the distillation process, requiring no additional generation costs.
2) An adaptive Prototype Sampling (APS) module is proposed to alleviate the domain shift issue between the unlabeled data and the original data.

**Limitations:**

Some concerns are listed below:
1) The adaptive prototype sampling module is proposed to obtain the substitution data with a more similar distribution to the original data. It is unclear why three score indicators (i.e., sc_i, so_i and sd_i) contribute equally to the total score (S_total), which should be further discussed. If not, which of them is more critical to the sampling process?
2) Missing some crucial literatures, such as
-- Learning to Retain while Acquiring: Combating Distribution-Shift in Adversarial Data-Free Knowledge Distillation, CVPR 2023.
-- Unpacking the Gap Box Against Data-Free Knowledge Distillation, T-PAMI 2024.

**Suitability:**

2

---

### Official Review · Reviewer_o8a6 · 2024-05-24

**Rating:** 2
**Confidence:** 3

**Summary:**

The paper introduces a framework named ODSD for Data-Free Knowledge Distillation, dedicated to addressing the issue that existing methods mostly rely on designing additional generative modules to synthesize substitute data, leading to high computational costs. It also points out the neglect of the abundant, easily accessible, low-cost, unlabelled open-world data. Through an adaptive sampling module, ODSD samples data from the open world close to the original data distribution and introduces low-noise representations to mitigate domain shift problems. It then constructs the structured relationships of data examples to utilize the knowledge in the data through the student model itself and the structured representations of the teacher. The method has shown notable improvements over various comparative baselines.

**Strengths:**

1. The comparative experiments in the article employ numerous baselines, making the comparisons comprehensive.
2. The article demonstrates the computational efficiency of the method, significantly reducing the computational cost compared to generative approaches.
3. The visual results presented in the article are intuitive and demonstrate the effectiveness of the method in semantic segmentation tasks.

**Limitations:**

1. The notation and formulaic expressions in the article could be improved. For example, the cosine similarity is directly denoted as cos(), which is misunderstood as a common trigonometric function and does not follow the conventional notation in the field of contrastive learning, such as using 'cos_sim' or 'sim'. Despite 'C' and 'K' being constants, the lowercase 'c' and 'k' are used to denote variables, and these are also employed as pronouns for prototype names. Additionally, the initial generation of prototypes mentions "is clustered [24]", leaving the reader uncertain about the actual process of obtaining prototypes.
2. The motivation of the paper is not well substantiated. The premise is that current methods overlook the massive amounts of easily accessible, low-cost, unlabeled open-world data. However, the distributional differences between datasets like ImageNet, CIFAR, and Flickr1M are minimal, or the paper should have provided statistics to display the distribution before and after selection. There is a lack of experiments with more diverse unlabeled open-world data to support this motivation. Furthermore, closely related distributions do not validate the effectiveness of the proposed DCRD module, a point not demonstrated by the ablation experiments.
3. The referencing style could be more appropriate. Expressions such as "the definition [22]," "are clustered [24]," and "with less noise interference [1]" are seldom used. Specific work names should be employed to provide clearer context. The expression of task definition in the summary is also lacking.
4. The structured relation repeatedly mentioned in the paper is not well defined, and it is also confusing why the relationship between the organizational structure of data can be easily learned through comparison learning.

**Suitability:**

3

---

### Official Review · Reviewer_5xbL · 2024-06-07

**Rating:** 4
**Confidence:** 2

**Summary:**

The paper proposes a novel approach to Data-Free Knowledge Distillation (DFKD) using an Open-world Data Sampling Distillation (ODSD) method. The approach aims to train student models effectively without the original training data or costly generative models. This method includes Adaptive Prototype Sampling (APS) to sample unlabeled data similar to the original data distribution and Denoising Contrastive Relational Distillation (DCRD) to reduce noise and capture data relationships. Experiments on CIFAR-10, CIFAR-100, NYUv2, and ImageNet datasets show state-of-the-art performance with reduced computational costs.

**Strengths:**

* The methodology is clearly explained,and the inclusion of figures helps in understanding the proposed method.
* The paper provides comprehensive experimental results on multiple datasets (CIFAR-10, CIFAR-100, NYUv2, and ImageNet). The results show significant improvements in accuracy and efficiency, supporting the claims of the proposed method's effectiveness.

**Limitations:**

Potentially Unfair Performance Comparison: The performance comparisons in the paper might not be entirely fair, as the time required for the teacher model to make predictions on the entire unlabeled open-world dataset is not accounted for. Additionally, the method involves sampling 150k or more examples for CIFAR-10/100, which far exceeds the original dataset sizes. The authors could provide a comparison of training hours to address this concern.
Generalization to Rare Image Categories: The datasets used, such as CIFAR and ImageNet, consist of common image categories, making it easier to sample relevant training data from an open-world dataset. It is uncertain whether the proposed method would perform as well on tasks involving rare image categories.

**Suitability:**

3

---

### Official Review · Reviewer_aeKW · 2024-06-08

**Rating:** 5
**Confidence:** 2

**Summary:**

The paper "Sampling to Distill: Knowledge Transfer from Open-World Data" introduces a novel method called Open-world Data Sampling Distillation (ODSD) aimed at enhancing Data-Free Knowledge Distillation (DFKD). The motivation behind this work is to eliminate the reliance on computationally expensive generation modules and to leverage the vast amounts of readily available, low-cost, unlabeled open-world data while addressing the domain shift issues inherent in existing methods. Key contributions include the Adaptive Prototype Sampling (APS) module, which samples data closely resembling the original dataset to mitigate domain shifts, and the Denoising Contrastive Relational Distillation (DCRD) module, which builds structured relationships among multiple data examples to improve knowledge transfer. The ODSD method improves upon Chen et al.'s DFND by reducing computational costs and enhancing accuracy, achieving significant performance gains across various datasets such as CIFAR-100 and ImageNet, demonstrating improvements in accuracy on ImageNet.

**Strengths:**

1. Theoretical Approach: The Adaptive Prototype Sampling (APS) and Denoising Contrastive Relational Distillation (DCRD) modules are theoretically robust. APS mitigates domain shift by sampling data that closely matches the original distribution, while DCRD enhances knowledge transfer by building structured relationships among multiple data samples.

2. Technical Correctness: The methodology is sound, with detailed explanations and justifications for APS and DCRD. The use of cosine similarity for outlier detection and structured relational distillation is well-founded, ensuring logical consistency and technical rigor.

3. Adequate Improvement and Evaluation: The paper provides extensive experiments on benchmark datasets. These evaluations compare ODSD against existing methods, showing significant improvements in accuracy and efficiency, with up to 9.59% accuracy gains on ImageNet.

4. Clarity: The paper is well-organized and clearly written, with explicit motivations, methodologies, and contributions. Detailed diagrams and tables enhance understanding, and the logical structure ensures accessibility for readers.

**Limitations:**

1. Novelty: While the proposed ODSD method introduces a sampling-based approach to DFKD, the concept of using open-world data for knowledge distillation is not entirely new. Previous works, such as Chen et al.'s DFND, have explored sampling-based methods. The primary novelty lies in the specific implementation of APS and DCRD, but the overall approach may be seen as an incremental improvement rather than a groundbreaking innovation.

2. Limited Exploration of Different Domains: Although the method is tested on several datasets, its effectiveness across a wider range of domains and tasks is not fully explored. Additional experiments on more diverse datasets would strengthen the generalizability claims.

**Suitability:**

2

---

### Meta-Review · Area_Chair_3QQg · 2024-07-01

**Recommendation:** Accept (Poster)
**Confidence:** 4

**Metareview:**

Thanks for authors's rebuttal and reviewers' contributions. All the reviews are positive and the work is accepted by MM.